# Study of Stress Distribution Characteristics of Reinforced Earth Retaining Walls under Cyclic Loading

**He Wang** [1,2,*], **Jian Ma** [1], **Guangqing Yang** [1,2] **and Nan Wang** [1]

1   School of Civil Engineering, Shijiazhuang Tiedao University, Shijiazhuang 050043, China
2   State Key Laboratory of Mechanical Behavior and System Safety of Traffic Engineering Structures, Shijiazhuang Tiedao University, Shijiazhuang 050043, China
*   Correspondence: wanghe@stdu.edu.cn

**Abstract:** The stress-distribution angle is an important parameter for the design of retaining walls and foundation beds and has a non-negligible role in the rationality of engineering design. There is a lack of research on stress distribution in reinforced earth-retaining walls under cyclic loading. In order to study the stress distribution characteristics of geogrid-reinforced soil-retaining walls (GRSW) under cyclic loading, the stress distribution characteristics of GRSW under a different number of load cycles were analyzed by field tests, and the effects of the length of reinforcement, the friction coefficient of the reinforcement–soil interface and the modulus of reinforcement on the stress distribution characteristics of GRSW were analyzed by numerical simulation. The results show that, with the increase in the number of load cycles, the vertical dynamic earth pressure shows a noticeable decreasing trend from high to low along the wall height. The vertical dynamic earth pressure increases first and then decreases along the length of reinforcement. When the number of load cycles increases to more than 100,000, the stress-distribution angle of the GRSW does not change much, the upper part remains at 35~79°, and the middle part remains at 47~68°. The influence depth of stress distribution in GRSW is about 1.13 times the wall height. The interfacial friction coefficient of reinforced soil has a superior influence on the stress distribution in GRSW, followed by the length and modulus of reinforcement.

**Keywords:** geogrid-reinforced soil-retaining wall; stress distribution; field test; cyclic load; numerical simulation

## 1. Introduction

With the rapid development of the social economy, both technology and theoretical research, reinforced soil-retaining walls have gained extraordinary development. In engineering, the stability of the retaining wall is the primary consideration, and the stress distribution caused by the external load is the key factor affecting the stability of the reinforced soil-retaining wall.

At present, domestic and foreign scholars have done some research on the internal stress-distribution characteristics of reinforced soil structures caused by an external load. Zhou Fen et al. [1] carried out a model test with a load applied on top of the unreinforced zone to analyze the distribution of subgrade soil pressure and its development pattern during the construction and loading phases, and the distribution of the incremental subgrade soil pressure caused by the load indicated that the reinforced zone could effectively block the distribution of vertical soil pressure. Wang Jiaquan et al. [2] studied the effect of the change in the length of the grid and the number of longitudinal and transverse ribs on the bearing performance of reinforced earth-retaining walls by conducting indoor model tests of reinforced retaining walls. Fattah et al. [3] investigated the effect of geogrid reinforcement on dynamic load transfer in underground structures through model tests and found that the pressure above the tunnel crown was reduced by 13–65% when geogrid reinforcement was used. Liao Zhiyong et al. [4] determined through an in-situ load test that

the subgrade surface load mainly affects the distribution of earth pressure on the upper part of the retaining walls and that the corresponding lateral earth pressure shows an approximately inverted triangle distribution. Due to the influence of dispersion and the unloading arch effect of reinforced soil, the vertical earth pressure decays faster than in the traditional retaining wall. Wang He et al. [5] analyzed the stress distribution characteristics of the geogrid-rewrapped reinforced earth-retaining wall under construction and wall top loading based on model tests and determined that the stress-distribution angle of the upper part of the reinforced earth-retaining wall increases and then decreases with the increase of external loading, that there is a maximum stress-distribution angle, and that the stress-distribution angle of the reinforced reinforcement is measured to be around 50°. Wang He et al. [6] measured the structural behavior of a geogrid-reinforced retaining wall (GRSW) with a deformation buffer zone (DBZ) under a static load by model tests and numerical simulations and determined that the reinforced earth-retaining wall with a DBZ structure can reduce the stress-energy release of the external load. DEMIR A et al. [7] and RAJESH S et al. [8], by conducting large-scale field tests and numerical simulations, concluded that horizontal reinforcement increases the stress-distribution angle of the reinforcement bedding, the dimensions of which are influenced by the nature and compaction of the soil between the layers, the strength, and the spacing of the reinforcement. Wang et al. [9] demonstrated that he dynamic load of lower amplitude makes the reinforced soil structure more compact through vibration, that the stress in the soil is mainly concentrated in the middle of the retaining wall, that the bearing capacity of the GRS retaining wall is increased, that high-amplitude dynamic loads weaken the embedded locking effect between the geogrid and the soil, and that the bearing capacity of the GRS retaining wall is reduced. In numerical simulation, Lu Yonggang [10] determined through finite element simulation analysis that the stress-distribution angle of the reinforced soil-retaining wall is the main change between 38~70°; the lower part of upper wall in the stress-distribution angle is bigger and smaller, and the stress-distribution angle decreases with the increase in the vertical spacing of ribbon, with the ribbon length of the pile, the strength of the ribbon, and driving speed increases. Zhou Hongwei et al. [11] studied the mechanism of stress distribution and settlement under the strip loading of reinforced foundations in subducted soil caverns through large-deformation finite element analysis. The stress concentration area between layers will be incomplete; for the same reinforcement range, increasing the number of layers has less effect on stress distribution; increasing the reinforcement range is more helpful for stress diffusion. The above studies show that, after laying reinforcing materials with a certain tensile strength horizontally in the soil, they are able to change the stress-distribution behavior in the earth caused by external loads based on the reinforcing soil interaction [12–14].

Stress distribution in a reinforced earth-retaining wall caused by the external load is the direct cause of the deformation of reinforced earth-retaining walls, and if the stress is too large, the wall is prone to instability damage. Compared with unreinforced soil-retaining walls, the reinforcing effect of reinforcing materials in reinforced soil-retaining walls affects stress distribution. The soil particles in the retaining walls are prone to deflection and movement under cyclic loading, which changes the stress-distribution behavior. Currently, the research on stress distribution in reinforced soil structure mainly focuses on reinforced soil mats. Nevertheless, the research on stress distribution in reinforced soil-retaining walls under cyclic loading is insufficient; in particular, the research on stress distribution characteristics of reinforced soil-retaining walls through field tests is rarely reported; moreover, the stress-distribution angle is also an essential parameter for retaining wall and bed design, and the change in the stress distribution has a significant influence on the design of a retaining wall and bed. Therefore, it is necessary to study the stress distribution characteristics of reinforced earth-retaining walls under cyclic loading. To this end, this paper investigates the stress distribution characteristics of reinforced soil-retaining walls under cyclic loading and analyzes the influencing factors based on the intercity

railroad project between Qingdao and Rongcheng using a combination of field tests and numerical simulations.

## 2. Materials Methods

The field-test section is the section DK315+913 of the Qingdao to Rongcheng Intercity Railway. The left and right sides of the section adopt modular GRSW. The right reinforced earth-retaining wall is selected for earth pressure monitoring. The reinforcing material is a unidirectional high-density polyethylene geogrid (HDPE), model EG130R and EG170R. The geogrid material parameters are shown in Table 1, and a total of 26 layers of reinforcing material are laid. The GRSW filling material is cohesive, and its parameters as shown in Table 2. The foundation is the CFG pile composite foundation; the panel is a C30 concrete precast block, and the module shape size is 0.5 m × 0.3 m × 0.3 m. The test section type and monitoring element arrangement are shown in Figure 1.

**Table 1.** Main mechanical properties of geogrids [15].

| Material Type | Tensile Modulus/(kN·m$^{-1}$) | Strength at 2% Strain/(kN·m$^{-1}$) | Strength at 5% Strain/(kN·m$^{-1}$) | Tensile Strength/(kN·m$^{-1}$) |
|---|---|---|---|---|
| EG130R | 1825 | 36.5 | 72 | 130.6 |
| EG170R | 2625 | 52.5 | 103 | 197.4 |

**Table 2.** Main mechanical properties of the filling.

| Material | Modulus of Elasticity (kN/m$^2$) | Cohesion (kN/m$^2$) | Natural Unit Weight kN/m$^3$ | Internal Friction Angle (°) |
|---|---|---|---|---|
| Filling | $5 \times 10^4$ | 30 | 18 | 35 |

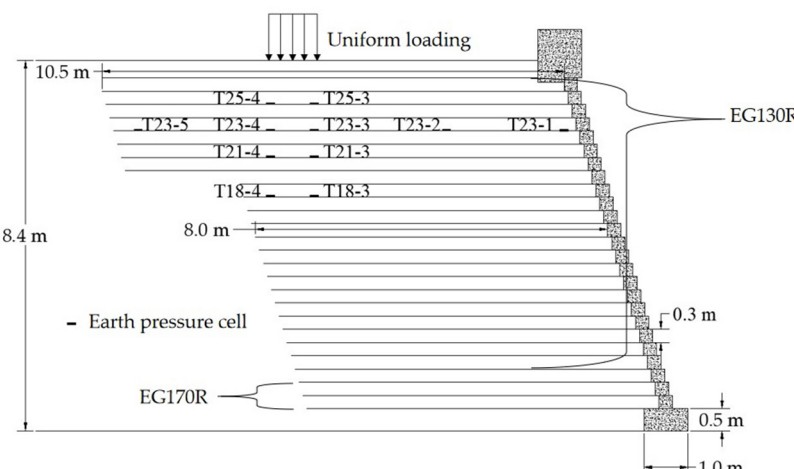

**Figure 1.** Arrangement of test elements in GRSW with a concrete block face located in the right side of section DK315+913.

The in situ excitation test system was used to apply cyclic loads on site, as shown in Figure 2. The cyclic load frequency is 13 Hz; the load cycle times are 1,000,000 times, and the load is 37 kPa. Based on the dynamic stress variation range of the subgrade surface in the field test, the cyclic load amplitude is determined to be 13.56 kPa, and the peak value is 80.62 kPa, as shown in Table 3. During the cyclic load application, the mechanical properties of the GREW were monitored in real-time, and Figure 3 shows some of the measured vertical dynamic earth pressure–time curves. This text analyzed the vertical earth pressure data when the number of load cycles were 100,000; 300,000; 500,000; 700,000; and 1,000,000.

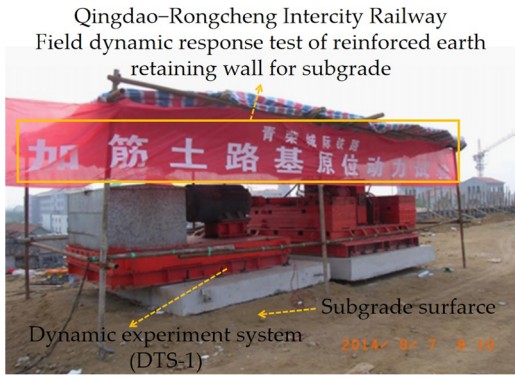

**Figure 2.** Picture of the vibration test at the site.

**Table 3.** Range of the dynamic load.

| Loading Frequency/Hz | Maximum Dynamic Stress/kPa | Minimum Dynamic Stress/kPa | Dynamic Stress Amplitude/kPa |
|:---:|:---:|:---:|:---:|
| 13 | 80.62 | 53.50 | 13.56 |

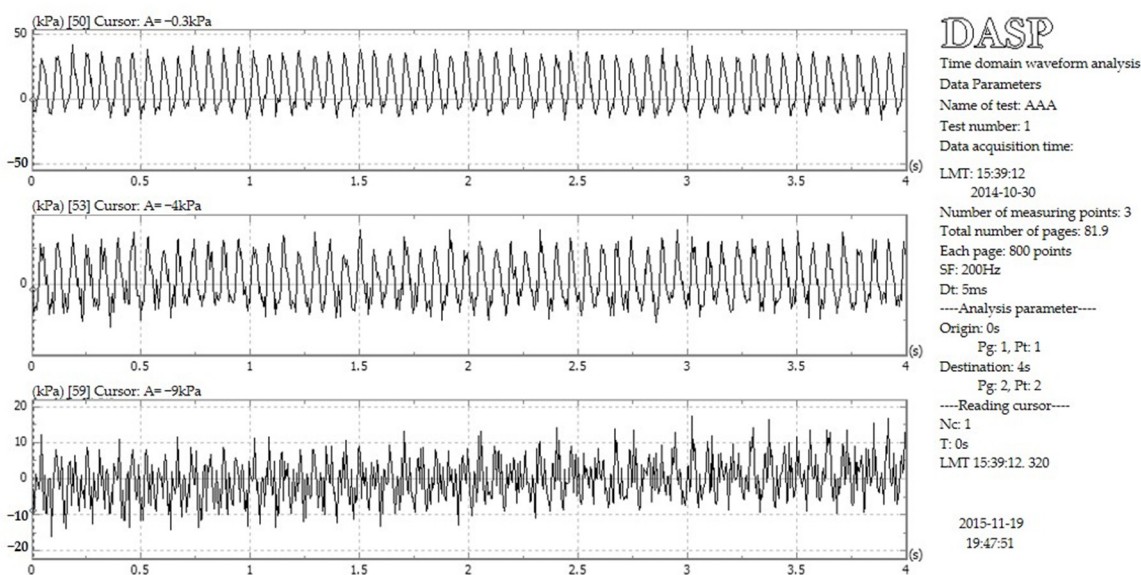

**Figure 3.** Time–history curve of the measured vertical dynamic soil pressure.

## 3. Results and Discussion

### 3.1. Distribution of Vertical Earth Pressure

Figure 4 shows the variation of the vertical dynamic earth pressure at each point in the GRSW with the gradually increasing loading. From the figure, it can be seen that, with the increase in loading times, the increase in the vertical dynamic earth pressure at the upper part of the GRSW on the right side of the DK315+913 section is relatively significant and then tends to be stable. This situation should be due to the lack of compactness caused by the original stable structure under the action of the excitation force being destroyed. When the soil particles are rearranged, the soil compactness increases further, and a stable structure is formed several times again. The lower part of the DK315+913 retaining wall has a higher compaction degree and a more substantial reinforcement material reinforcement effect. The compaction degree of the retaining wall reaches a higher degree, so it is difficult to destroy the original stable structure when it is subjected to vibration again. Hence, its vertical dynamic earth pressure has almost no change.

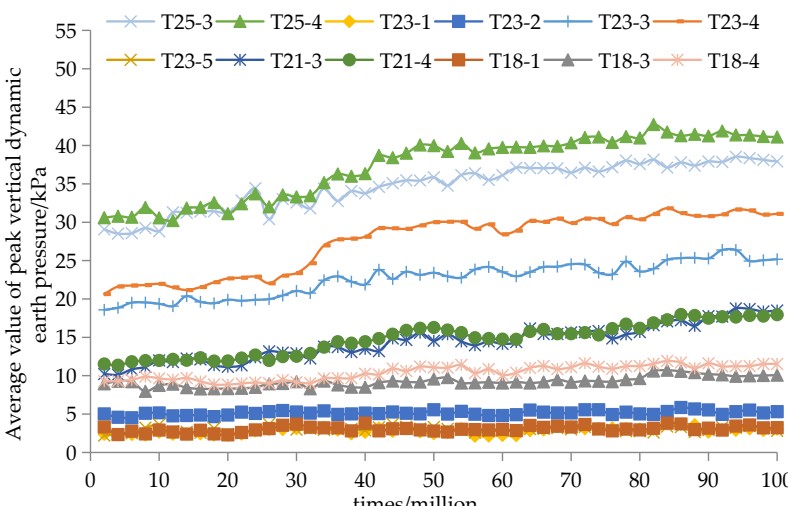

**Figure 4.** Distribution of vertical dynamic soil pressure with increases in load cycle number.

Figure 5 shows the distribution of the peak vertical dynamic earth pressure along the wall height of the GRSW at different load cycles in the longitudinal section where the earth pressure box T25-4 is located, calculated from the field-test monitoring data. It can be seen from the figure that the vertical earth pressure after loading is nonlinear along the wall height. Along the wall height, the decay trend from high to low is presented, the attenuation rate is lower and smaller, and the decay rate is 83.5%~86.9% in the wall height range of 2.7 m. The reason for the analysis was that there were gaps between the soil particles, and they were not sufficiently compacted. Under the action of its own gravity and external load, the pores in the upper part of the soil inside the retaining wall are reduced, the soil becomes dense, and the energy of the force is gradually consumed through the friction between the soil particles and other damping effects. As the soil becomes compact, the force energy is gradually consumed by damping effects such as friction between soil particles. At different heights of retaining walls, with the increase of load cycles, the vertical earth pressure also increases. The increase rate is the smallest at a depth of 2.7 m from the wall top, and the increase rate is the largest at a depth of 1.8 m from the wall top, which are 19.3% and 49.7%, respectively.

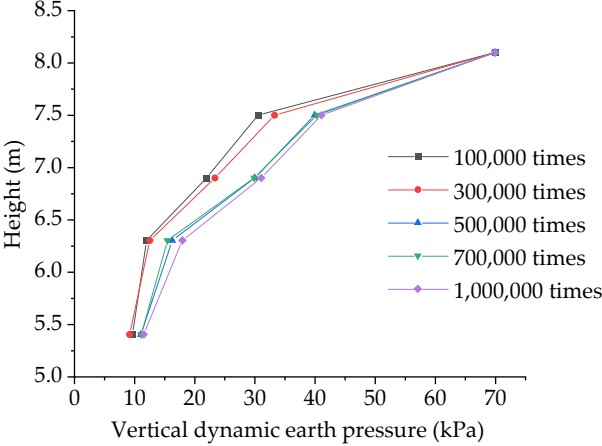

**Figure 5.** Distribution of vertical earth pressure along the wall height.

Figure 6 shows the distribution of the peak vertical dynamic earth pressure along the length of the reinforcement at a wall height of 6.9 m for different numbers of load cycles. According to the figure, the vertical dynamic earth pressure varies nonlinearly along the length of the reinforcement and attends to increase and then decrease with the increase of

the distance from the wall panel, and the peak value appears at the middle and rear of the reinforcement. As the number of load cycles increases, the vertical dynamic earth pressure near the wall panel is almost unaffected. This is due to the horizontal displacement of the wall surface, which reduces the vertical dynamic earth pressure near the wall surface by releasing it. The vertical earth pressure in the rear of the geogrid increases gradually and tends to be stable with the increase of load cycles. This is due to the gradual densification of the soil under the external load and the gradual attainment of the maximum densification at that load level.

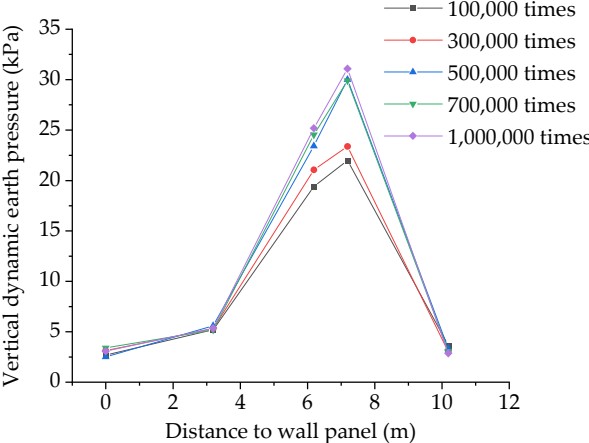

**Figure 6.** Distribution of vertical peak earth pressure along the length of reinforcement at a wall height of 6.9 m.

### 3.2. Stress Distribution Characteristics

The angle between the boundary line formed by the points with zero vertical stress on both sides of the retaining wall and the vertical direction is the stress-distribution angle. The stress-distribution angle can more intuitively reflect the influence range of the external load on GRSW. Since it is difficult to obtain the magnitude of the stress-distribution angle directly by calculation in practical engineering, the triangular linear interpolation method is used to calculate the coordinates of the points with zero stress at different wall heights by the center-of-gravity coordinate formula and connect them to obtain the stress-distribution range of the retaining wall.

The principle of triangle linear interpolation is that, given a triangle vertex coordinates and values, as shown in Figure 7, the weight influence of each vertex on a specific point inside the triangle is calculated by the center-of-gravity coordinate formula. Then the coordinates and values of a specific point inside the triangle are obtained. See Equations (1)–(3). In this paper, assuming that the value of the stress zero point is known, the coordinates of the additive stress zero point can be obtained by the trial algorithm.

$$P(x) = (1 - m - n) \times P1(x) + m \times P2(x) + n \times P3(x) \tag{1}$$

$$P(y) = (1 - m - n) \times P1(y) + m \times P2(y) + n \times P3(y) \tag{2}$$

$$P = (1 - m - n) \times P1 + m \times P2 + n \times P3 \tag{3}$$

where: $1 - m - n$ is the weight of point P1, m is the weight of point P2, and n is the weight of point P3.

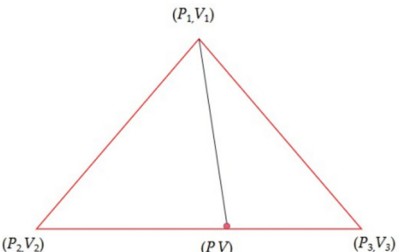

**Figure 7.** Triangle linear interpolation diagram.

Due to the small number of dynamic earth pressure boxes placed in the field test, it is not enough to obtain the stress-distribution law of the retaining wall by this method. To solve this problem, according to the vertical earth pressure value obtained from field monitoring, a contour map of vertical earth pressure was drawn using drawing software. Take the vertical earth pressure at 300,000 load cycles, as shown in Figure 8, to obtain the required coordinates and the value of earth pressure. Then the calculation is performed by linear interpolation inside the triangle, and finally, the stress distribution line is premeditated, as shown in Figure 9.

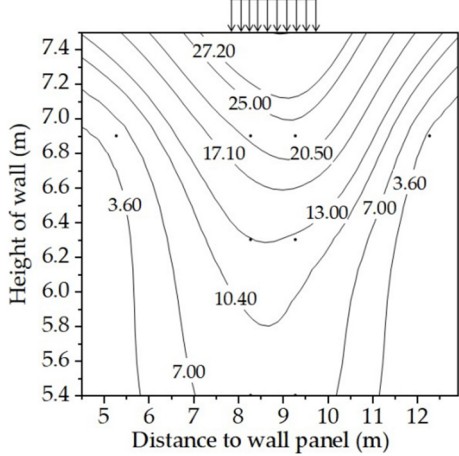

**Figure 8.** Contour diagram of vertical dynamic earth pressure in the retaining wall when the number of load cycles is 300,000.

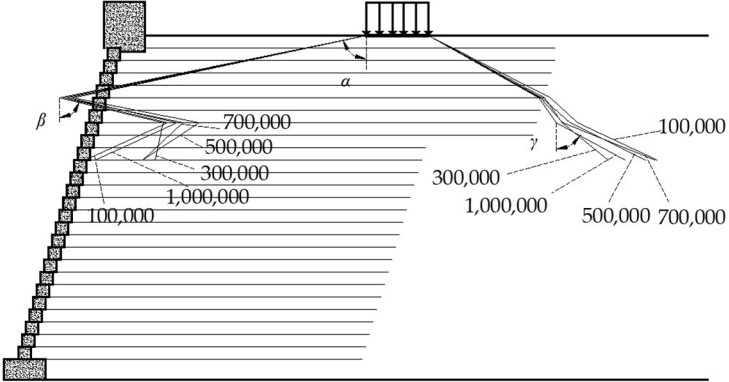

**Figure 9.** Diagram of the measured stress distribution of GRSW.

Figure 9 shows the stress-distribution range in the GRSW caused by the top cyclic load under different load cycles. The specific distribution angle is shown in Table 4. It can be seen that the stress caused by an external load is gradually attenuated from high to low, so the stress-distribution range is gradually expanded and then gradually contracted. The

difference between positive and negative values is caused by the dispersion and decay of stress. The stress-distribution angle of the upper retaining wall ranges from 35° to 79°, which is larger than that of the unstiffened body. Under the interaction of reinforced soil, the reinforced soil is denser, the stress-distribution range is more extensive in the upper part of the retaining wall, and more soil bears the stress, so the stress decays faster, and the upper part of the retaining wall bears most of the stress. The influence range of the stress on the wall panel is concentrated in the upper part, so the upper part of the GRSW is prone to sizeable horizontal deformation.

**Table 4.** Stress-distribution angle of reinforced soil.

| Location | Number of Loading Cycles | Distribution Angle $\alpha$ (°) | Distribution Angle $\beta$ (°) | Distribution Angle $\gamma$ (°) |
|---|---|---|---|---|
| Near wall panel side | 100,000 | 78 | −77 | 66 |
| | 300,000 | 78 | −77 | 13 |
| | 500,000 | 78 | −79 | 57 |
| | 700,000 | 78 | −78 | 47 |
| | 1,000,000 | 78 | −77 | 66 |
| Away from wall panel side | 100,000 | 63 | 47 | 65 |
| | 300,000 | 62 | 35 | 51 |
| | 500,000 | 61 | 49 | 64 |
| | 700,000 | 61 | 42 | 68 |
| | 1,000,000 | 61 | 36 | 61 |

On the side near the wall panel, the stress-distribution angle at the upper part of the retaining wall is the same as that at the side away from the wall panel, but there will be noticeable fluctuation at the middle and upper parts. It may be that the vertical soil pressure is redistributed due to the horizontal displacement of the wall panel, and the stress-distribution angle does not exhibit apparent regularity phenomenon. The stress-distribution angle of the middle and upper part of the GRSW is smaller than that of the upper part.

During the construction design of the retaining wall, the subgrade bed will often be designed within the upper part of the retaining wall. The variation of the stress-distribution angle within the retaining wall will have an impact on the design of the retaining wall and the roadbed.

In the design of retaining walls, Federal Highway Administration stipulates that the stress-distribution angle generated by external loads is 26.6°. As the stress-distribution angle increases, the dynamic stress generated by external loads decreases and the effective length of the tendons increases, so the length of the tendons can be reduced appropriately under the premise of meeting the design specifications. In bed design, American Railway Engineering and Maintenance-of-Way Association regulations require that the stress acting on the lower fill is less than its allowable stress. In the case of a reinforced bed, with the increase of the stress-distribution angle, the decay of dynamic stresses within the bed increases and the dynamic stresses become smaller, so the bed thickness can be reduced appropriately in the design. Some detailed studies on this aspect will be conducted later.

### 4. Numerical Simulation

PLAXIS finite element software was used for modeling. PLAXIS is capable of simulating complex engineering geological conditions and is especially suitable for deformation and stability analysis. Compared with other FE models, it has the advantages of simple operation, high accuracy, and a small amount of computational time consumption, and the simulated structure can be well matched with the test curve.

### 4.1. Finite Element Model

Plaxis8.5 finite element software was used for modeling. The GRSW was 8.4 m high, and the GRSW was filled with viscous soil. The reinforcement length was 8~10.5 m, and the vertical spacing was 0.3 m. The foundation was 10 m thick; the upper layer was 5 m silty clay; the lower layer was 5 m bedrock; the foundation was treated by CFG concrete pile composite foundation; the pile length was 4.4 m; the pile diameter was 0.4 m; the pile spacing was 1.8–2.2 m; C30 concrete was used to pour a 0.3 m capstone on top of the pile, and a 0.5 m-reinforced gravel cushion was laid on top of the capstone. The 15-node plane strain model was adopted, and the numerical model is shown in Figure 10.

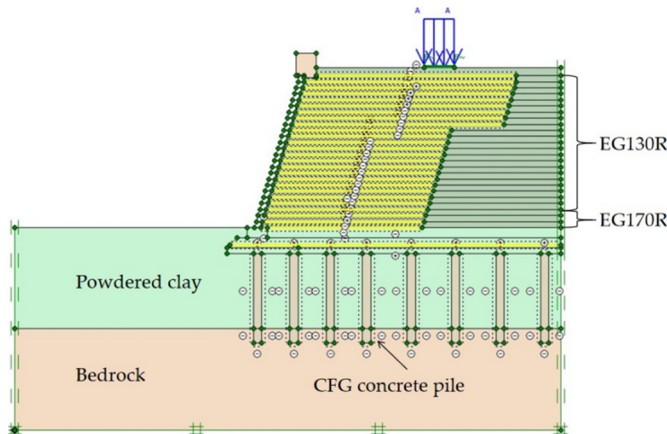

**Figure 10.** Finite element model diagram of a DK315+913 reinforced soil-retaining wall.

Considering the influence of reinforcement–soil interaction, the interface unit defines the contact surface of the reinforcement and soil. In the model, the interaction degree of the soil is reflected by the friction coefficient of the soil interface. The friction coefficient Rinter of the reinforced soil interface was 0.67. The boundary conditions are horizontal constraints on both sides and horizontal and vertical constraints at the bottom of the foundation. The principal structure models and parameters used for each material are shown in Table 5.

**Table 5.** Parameters of geotechnical and concrete materials.

| Name | Material MODEL | Modulus of Elasticity (kN/m$^2$) | Cohesion (kN/m$^2$) | Natural Unit Weight kN/m$^3$ | Internal Friction Angle (°) |
|---|---|---|---|---|---|
| Bedrock | Line elasticity | $1.2 \times 10^7$ | — | 27 | — |
| Powdered clay | Mohr–Coulomb | 8950 | 15 | 19 | 30 |
| Gravel matting | Mohr–Coulomb | $5 \times 10^5$ | 1 | 20 | 40 |
| Pile cap | Line elasticity | $3.5 \times 10^7$ | — | 25 | — |
| Bold stone and foundation | Line elasticity | $3 \times 10^7$ | — | 24 | — |
| Filling | Mohr–Coulomb | $5 \times 10^4$ | 30 | 18 | 35 |
| CFG pile | Line elasticity | $3 \times 10^7$ | — | 25 | — |
| Panel | Line elasticity | $1.2 \times 10^7$ | — | 24 | — |

### 4.2. Numerical Model Validation

In order to test the rationality of parameter selection for the finite element numerical model, the numerical simulation value of the vertical ground pressure peak distribution rule at a height of 6.9 m at a load cycle number of 300,000 was selected for comparison and verification with the measured value in a field test, as shown in Figure 11. As can be seen from the figure, the numerical simulation value is basically consistent with the measured value in the field test. The slight difference between the two values is since the compaction of the fill throughout the retaining wall cannot be accurately controlled

in actual projects and may be uneven, whereas the compaction of the fill in the PLAXIS retaining-wall model is uniform by default. Uneven compaction will result in a weakening of the internal load-carrying capacity of the retaining wall and an increase in the stress-transfer capacity, resulting in larger field values for vertical stresses in the retaining wall than the simulated values. The error between the field measured value and the numerical simulation value is acceptable. Within a certain range, the coincidence degree is good, so the established model is reliable.

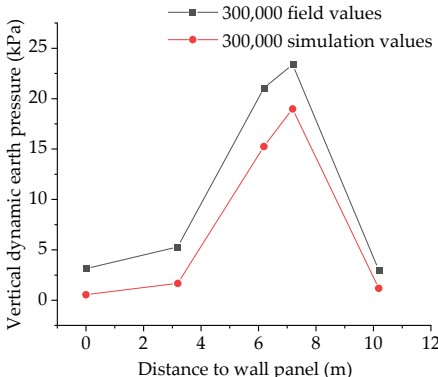

**Figure 11.** Distance from the back of the wall—vertical dynamic earth pressure comparison chart.

The simulation analysis of the load test by using PLAXIS shows that the PLAXIS simulation curve is in good agreement with the actual test curve. Although PLAXIS is a reasonable simulation of the test because the soil is a non-homogeneous body, its nature is randomly variable, so it still needs to accumulate and summarize the simulation of the real situation. With the continuous optimization of PLAXIS software and the accumulation of practical engineering experience, the application of PLAXIS in load tests will be more extensive.

## 5. Study on Distribution Characteristics of Additional Stress

It is difficult to obtain the distribution cloud of stress in numerical simulation, so the vertical stress fluctuation amplitude of each stress point at different heights in the retaining wall under a cyclic load is used to determine the position of stress zero. Under different loading, each stress point of gravity stress values are the same, so one should point to the fact that the smaller vertical-stress wave amplitude shows that the point is that the effect of cyclic loading is weak, and wave amplitude is small enough to ignore when suffering from a lack of timing; stress is almost zero, and the point of stress is zero. Because the vertical earth pressure at different heights of the retaining wall increases first and then decreases along the direction of the reinforcement, the stress amplitude of each stress point at different heights of the retaining wall was analyzed according to the above method, and the zero of stress was found by reducing the range of stress points, and the range of the distribution angle of stress in the retaining wall was obtained. Take the numerical model with the load cycle number of 300,000 times as an example, as shown in Figure 12, where the wall height is 7.5 m and the side near the wall panel is 1.7 m away from the wall panel.

According to the analysis of the above method, the stress-distribution law in the GRSW with different load cycles is obtained, as shown in Figure 13. It can be seen from the figure that the stress-distribution angle and the stress-distribution trend of the GRSW do not change at different load cycles, which is basically consistent with the field-measured results. Along the high direction of the wall, due to the displacement of the wall panel near the back of the wall, the stress-distribution angle in the middle of the retaining wall is small, and the stress-distribution angle in the upper part is 77°, and the stress-distribution angle far from the back of the wall is 72°. Away from the side of the wall panel, stress decays and collapses, probably because the soil close to the geotextile differs, with higher geotextile stiffness; the stress dispersion area decreases, and, as the soil is farther and farther away

from the geotextile, the stress of the soil mass transfer will become stable and positioned far away from the wall panel; the phenomenon of the stress decays, which, because of the model size, can cause what looks like a stress collapse. The stress-distribution angle in the upper part of the retaining wall is larger than that in the middle and lower parts. The distribution depth of stress in GRSW is about 1.13 times the wall height.

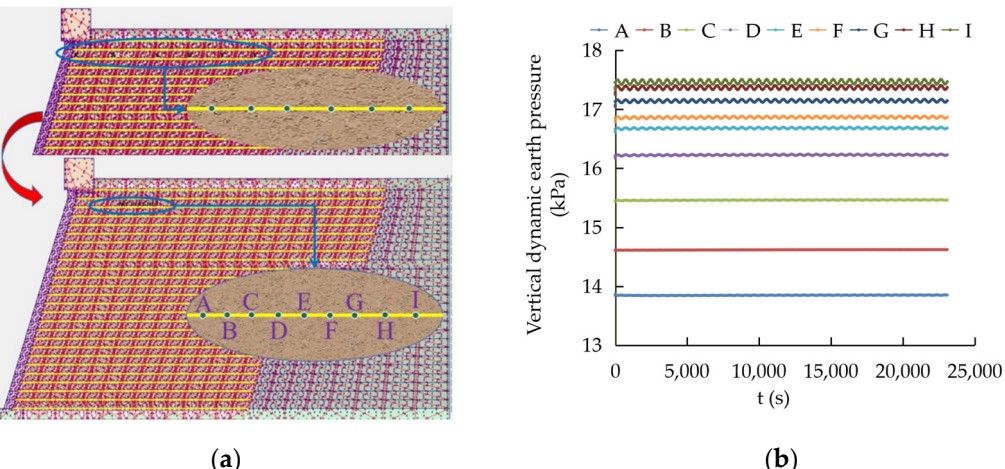

(**a**)  (**b**)

**Figure 12.** Flow chart for determining the zero point of stress; (**a**) selection process of stress points for the numerical model of retaining wall; (**b**) the vertical stress at each selected stress point.

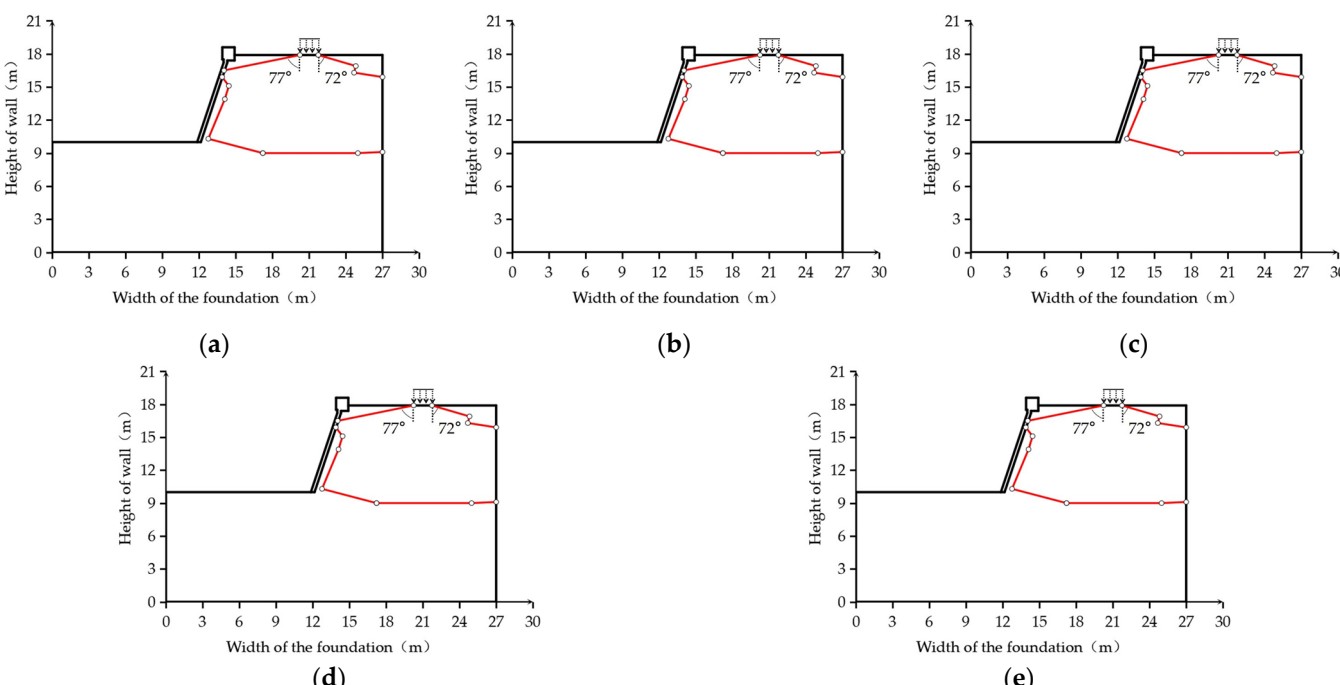

**Figure 13.** DK315+913 GRSW stress-distribution diagram; stress-distribution range of the retaining wall (red line): (**a**) under 100,000 cycles of loading; (**b**) under 300,000 cycles of loading; (**c**) under 500,000 cycles of loading; (**d**) under 700,000 cycles of loading; (**e**) under 1,000,000 cycles of loading.

## 6. Analysis of Influence Factors on Stress Distribution Characteristics of GRSW under Cyclic Loading

Numerical simulations are used to analyze the stress-distribution behavior in GRSW by varying the length of the reinforcement, the friction coefficient of the reinforcement–soil interface, and the modulus of the reinforcement, respectively. In turn, the influence of the main factors on the stress distribution characteristics of GRSW is investigated.

### 6.1. Influence of Interfacial Friction Coefficient of Reinforced Soil on the Stress Distribution Characteristics of GRSW

To study the influence of the friction coefficient between the reinforcement and soil on the stress distribution characteristics of GRSW, only the friction coefficient between reinforcement and soil was changed to observe the change law of the stress-distribution angle of GRSW under the condition that parameters such as the distance between reinforcement and reinforcement strength were unchanged. The friction coefficients of the stiffened soil interface were selected as 0.17, 0.27, 0.37, 0.47, 0.57, and 0.67, respectively, for comparative analysis at the time of 100,000 load cycles. According to Figure 14, the analysis results show that, with the increase of the friction coefficient of the stiffened soil interface, the stress-distribution range in the GRSW gradually increases, and the stress-distribution angle near the back side of the retaining wall are 60°, 72°, 74°, 74°, 75°, and 77°, respectively. The stress-distribution angle away from the back and upper retaining wall are 49°, 57°, 69°, 69°, 70°, and 72°, respectively. The reason is that the reinforced soil structure is strengthened, and the stress transfer range is more prominent due to the increase of friction, occlusion, and embedment between the reinforced soil and more soil participates in the stress caused by an external load.

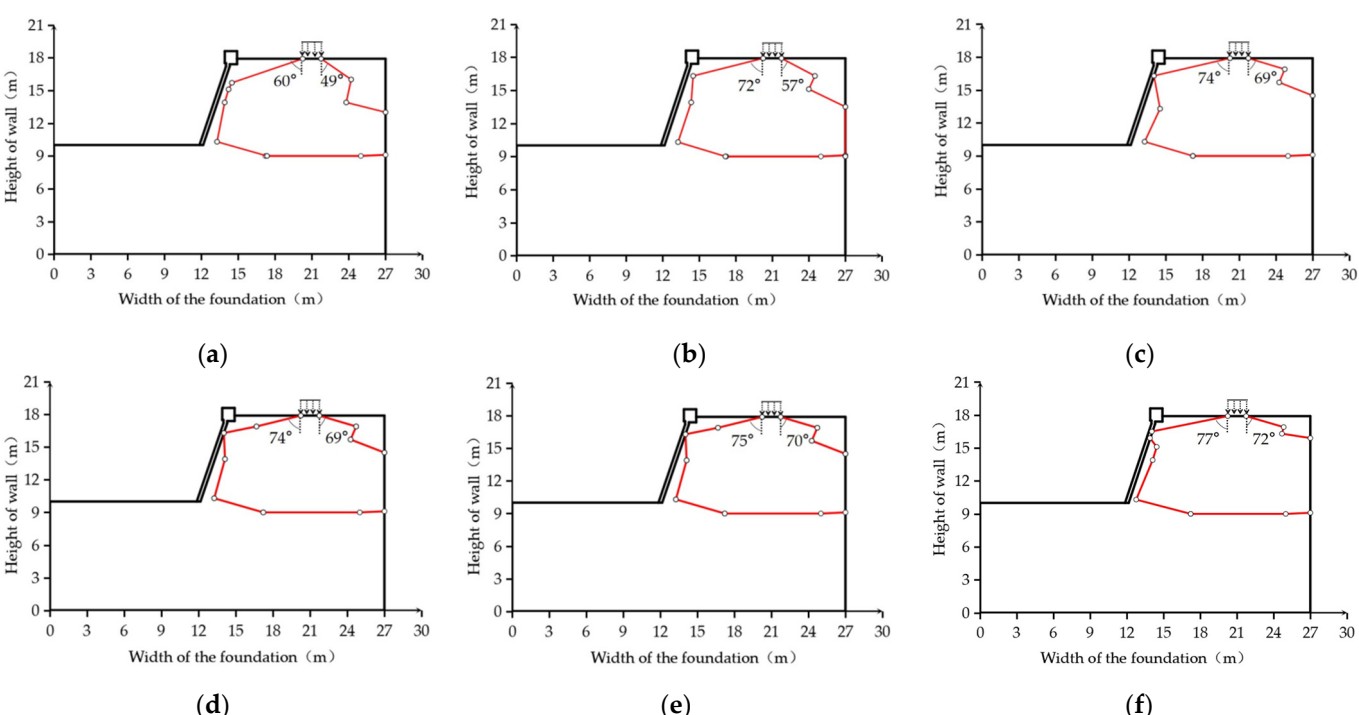

**Figure 14.** Stress distribution in reinforced soil-retaining walls with different friction coefficients at the reinforced soil interface; Retaining wall stress-distribution range (red line) when the friction coefficient of the reinforced soil interface is: (**a**) 0.17; (**b**) 0.27; (**c**) 0.37; (**d**) 0.47; (**e**) 0.57; (**f**) 0.67.

### 6.2. Influence of Reinforcement Modulus on Stress Distribution Characteristics of the Reinforced GRSW

The modulus of reinforcement was selected as 625 kN/m, 1025 kN/m, 2525 kN/m, and 3025 kN/m for comparative analysis, as shown in Figure 15. According to the results in the figure, the stress-distribution angle of the GRSW remains unchanged as the reinforcement modulus increases. The reason is that, although the increase in the modulus of the reinforcement will limit the lateral displacement and vertical settlement of the retaining wall, the stiffness of the reinforcement itself is substantial. The deformation of the retaining wall in service is minimal, so the change in the modulus of the reinforcement has little effect on the stress distribution of the retaining wall.

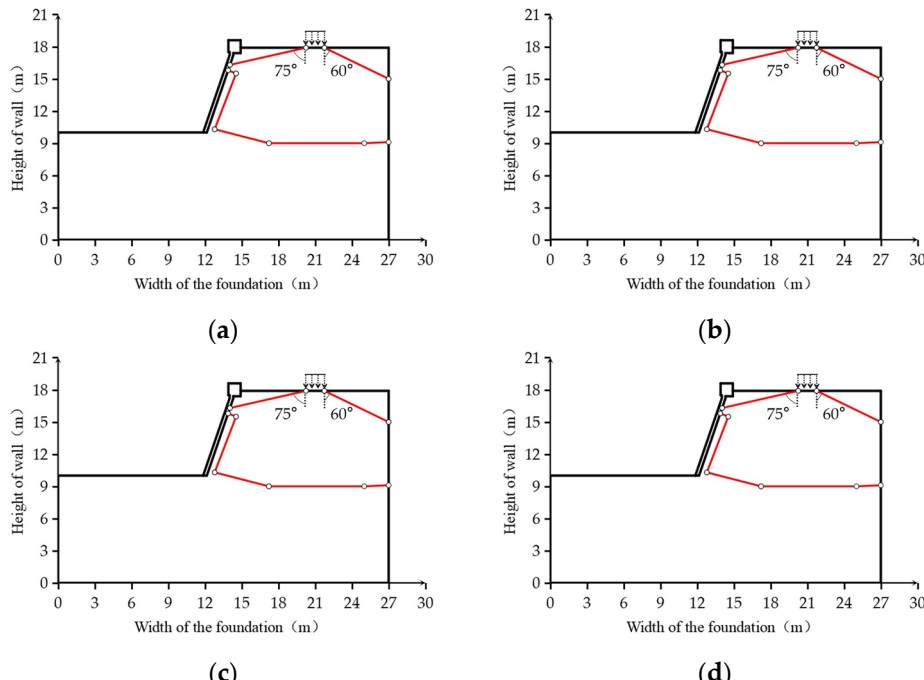

**Figure 15.** Stress distribution in GRSW with the different modulus of reinforcement parameters; Retaining wall stress-distribution range (red line) when the modulus of reinforcement is: (**a**) 625 kN/m; (**b**) 1025 kN/m; (**c**) 2525 kN/m; (**d**) 3025 kN/m.

*6.3. Effect of Reinforcement Length on the Stress Distribution Characteristics of GRSW*

Since the lengths of the reinforcement in the retaining wall in the field test were 10.5 m and 8 m, the lengths of the reinforcement in the numerical simulation analysis were selected as 8.5 m, 6 m, 9.5 m, 7 m, 12.5 m, and 10 m, respectively, and the stress distribution characteristics were compared at 100,000 load cycles, as shown in Figure 16. From the figure, it can be seen that, with the increase in the length of the reinforcement, the overall stress-distribution range within the GRSW is gradually expanding, and the stress-distribution angle increases, with the stress-distribution angles of 68°, 70°, 77°, and 77° in the upper part of the retaining wall near the back side of the wall and 64°, 70°, 72°, and 72° in the upper part of the retaining wall far from the back of the wall. The main reason for this is that the increase in the length of the reinforcement makes the reinforced area in the wall larger, which makes the distribution angle larger; however, when the length of the reinforcement is too long, the stress-distribution effect will not change much but will cause a waste of reinforcement.

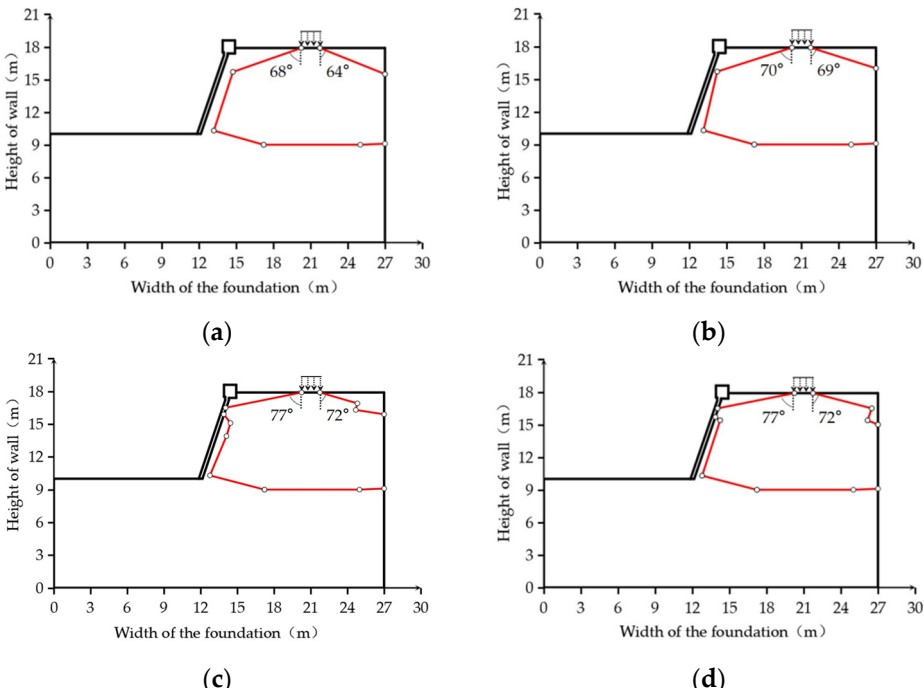

**Figure 16.** Stress distribution in the GRSW with different reinforcement length parameters; retaining wall stress-distribution range (red line) at (**a**) 8.5 m for long reinforcement and 6 m for short reinforcement; (**b**) 9.5 m for long reinforcement and 7 m for short reinforcement; (**c**) 10.5 m for long reinforcement and 8 m for short reinforcement; (**d**) 12.5 m for long reinforcement and 10 m for short reinforcement.

## 7. Discussion

According to the above research results, different designs of the reinforcement material will have different effects on the stress distribution of the reinforced soil-retaining wall. The influence of the friction coefficient of the ribbed soil interface is the largest, to ensure the full interaction between the ribs and soils, and the compaction degree should be guaranteed to the greatest extent during the construction of the project. The influence of the length of the reinforcement is greater than the modulus of the reinforcement, starting from the economy, in the design of the reinforcement soil-retaining wall, under the premise of safety and rationality, so you can choose a design scheme with a long length and low strength.

At present, there is very little research on the reinforcement optimization of the retaining wall under the premise of considering the stress distribution, and this paper is only a single study of the stress distribution from the aspect of the reinforcement, and then a comprehensive study and analysis will be carried out for this part.

## 8. Conclusions

(1) The vertical soil pressure inside the GRSW caused by external load gradually decays from high to low along the wall height and decays by 83.5%~86.9% within the range of 2.7 m of the wall height. The increase in the number of load cycles causes the vertical dynamic earth pressure to increase as well, with a minimum increase of 19.3% and a maximum increase of 49.7%.

(2) Along the tendon-length direction, the vertical dynamic soil pressure shows a nonlinear single-peak distribution, with the peak occurring at the middle and rear of the tendon; with the increase of load cycles, the vertical dynamic soil pressure near the wall panel is almost unaffected, and at the middle and rear of the tendon, the vertical dynamic soil pressure gradually increases and stabilizes with the increase of load cycles.

(3) The change in the number of load cycles has basically no effect on the stress-distribution angle of the GRSW; numerical simulation and field tests shows that the stress-

distribution angle is relatively large in the upper part of the retaining wall and smaller in the middle and lower part, and the stress angle is basically between 35° and 79°; the distribution influence depth of the stress in the GRSW is about 1.13 times the wall height.

(4) With the increase in the friction coefficient of the reinforced soil interface, the stress-distribution range within the GRSW gradually becomes larger; with the increase in the length of the reinforcement, the stress-distribution range within the GRSW is gradually expanded in general, and the stress-distribution angle also increases, but when the length of the reinforcement is too long, the stress-distribution effect basically does not change much; with the increase of the modulus of the reinforcement, the stress-distribution angle of the GRSW basically remains the same; Among the three influencing factors, the friction coefficient of the reinforced soil interface has the greatest influence on the stress-distribution characteristics of the reinforced soil-retaining wall, the length of the reinforcement material is second, and the change in the modulus of the reinforcement material has the most minor influence; the parameters of the reinforcement material should be selected reasonably in the construction of reinforced soil projects to avoid waste.

**Author Contributions:** Writing—original draft preparation, H.W.; writing—review and editing, J.M.; validation, G.Y. and N.W. All authors have read and agreed to the published version of the manuscript.

**Funding:** This research was supported by the National Natural Science Foundation of China (No. 52079078), the National Key Research and Development Program of China (No. 2022YFE0104600), and the Key Research and Development Plan of Hebei Province (No. 20375504D).

**Institutional Review Board Statement:** Not applicable.

**Informed Consent Statement:** Not applicable.

**Data Availability Statement:** The data used to support the findings of this study are available from the corresponding author upon request.

**Conflicts of Interest:** The authors declare no conflict of interest.

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
