# Peer review of "Study of Stress Distribution Characteristics of Reinforced Earth Retaining Walls under Cyclic Loading"

_applsci, doi:10.3390/app122010237_

Round 1
Reviewer 1 Report
Please see comments and suggestions in attached file

Author Response
亲爱的评论者
请参阅附件。

Reviewer 2 Report
Thank you for submitting your paper. The work done here draws attention to a significant subject characteristics of geogrid reinforceds oil retaining walls. I have found the paper to be interesting. However, several issues need to be addressed properly before the paper is being considered for publication. My comments including major and minor concerns are given below:
Please consider reviewing the abstract and highlight the novelty, major findings, and conclusions. I suggest reorganizing the abstract, highlighting the novelties introduced. The abstract should contain answers to the following questions:
What problem was studied and why is it important?
What methods were used?
What conclusions can be drawn from the results? (Please provide specific results and not generic ones).
The abstract must be improved. It should be expanded. Please use numbers or % terms to clearly shows us the results in your experimental work.
Please consider reporting on studies related to your work from mdpi journals.
The introduction is too short and authors need to expand it, mention in details past studies similar to this work, what they did and what were their main findings and critically evaluate their results against each then mention how does your work brings new knowledge and difference to the field.
Authors should remove all bulk citations unless they are given full credit.
The authors should remove all bulk citations, unless given full credit afterwards. The authors should check for this issue elsewhere in the manuscript.
Combine any small paragraphs of 5 lines or less with other paragraphs to improve the readability of the manuscript.
2. authors are encouraged to change “Project summary” to “materials and methods” or something more appropriate.
Table 1 needs a reference unless measured by authors
Figure 2 please remove or add arrows and text to clearly tell us what are we looking at in it.
Figure 3 is not clear and have poor resolution, please improve.
Combine Table 3 with previous tables if possible.
3. Test results and analysis change to Results and discussion
Lines 141-142 the authors need to support this claim with a reference(s)
Line 156 why it is unaffected? Authors need to provide stronger explanation for the observations they make
The title of the paper does not indicate there is simulation, authors need to reflect that in the title
In Figure 11, what are some of the reasons behind the discrepancy between the experimental and numerical results.
What are some limitation of your model, mention them properly.
What are differences between your FE model and previous ones?
Remove section 7 Discussion and merge with previous sections
Manuscript requires extensive formatting and tidying.
The results are merely described and is limited to comparing the experimental observation and describing results. The authors are encouraged to include a more detailed results and discussion section and critically discuss the observations from this investigation with existing literature.
Conclusion can be expanded or perhaps consider using bullet points (1-2 bullet points) from each of the subsections.
Author Response
亲爱的评论者
请参阅附件。

Round 2
Reviewer 2 Report
Dear authors,
Check line 393. something is no ok there (first sentence)
after that paper can be accepted.
